# Timing of Systemic Steroids and Neurodevelopmental Outcomes in Infants < 29 Weeks Gestation

**DOI:** 10.3390/children9111687

**Published:** 2022-11-03

**Authors:** Hemasree Kandraju, Bonny Jasani, Prakesh S. Shah, Paige T. Church, Thuy Mai Luu, Xiang Y. Ye, Miroslav Stavel, Amit Mukerji, Vibhuti Shah

**Affiliations:** 1Department of Paediatrics, Mount Sinai Hospital, Toronto, ON M5G 1X5, Canada; 2Division of Neonatology, Hospital for Sick Children, Toronto, ON M5G 1X8, Canada; 3Department of Newborn and Developmental Pediatrics, Sunnybrook Health Sciences Centre, Toronto, ON M4N 3M5, Canada; 4Department of Pediatrics, CHU Sainte-Justine, Montreal, QC H3T 1C5, Canada; 5Maternal-Infant Care Research Centre, Mount Sinai Hospital, Toronto, ON M5G 1X6, Canada; 6Neonatal Intensive Care Unit, Royal Columbian Hospital, New Westminster, BC V3L 3W7, Canada; 7Department of Pediatrics, McMaster University, Hamilton, ON L8S 3Z5, Canada

**Keywords:** preterm-infant, postnatal systemic steroids, mortality, neurodevelopment

## Abstract

Objective: To determine the association between postnatal age (PNA) at first administration of systemic postnatal steroids (sPNS) for bronchopulmonary dysplasia (BPD) and mortality or significant neurodevelopmental impairment (sNDI) at 18–24 months corrected age (CA) in infants < 29 weeks’ gestation. Methods: Data from the Canadian Neonatal Network and Canadian Neonatal Follow-up Network databases were used to conduct this retrospective cohort study. Infants exposed to sPNS for BPD after the 1st week of age were included and categorized into 8 groups based on the postnatal week of the exposure. The primary outcome was a composite of mortality or sNDI. A multivariable logistic regression model adjusting for potential confounders was used to determine the association between the sPNS and ND outcomes. Results: Of the 10,448 eligible infants, follow-up data were available for 6200 (59.3%) infants. The proportion of infants at first sPNS administration was: 8%, 17.5%, 23.1%, 18.7%, 12.6%, 8.3%, 5.8%, and 6% in the 2nd, 3rd, 4th, 5th, 6th, 7th, 8–9th, and ≥10th week of PNA respectively. No significant association between the timing of sPNS administration and the composite outcome of mortality or sNDI was observed. The odds of sNDI and Bayley-III motor composite < 70 increased by 1.5% (95% CI 0.4, 2.9%) and 2.6% (95% CI 0.9, 4.4%), respectively, with each one-week delay in the age of initiation of sPNS. Conclusions: No significant association was observed between the composite outcome of mortality or sNDI and PNA of sPNS. Among survivors, each week’s delay in initiation of sPNS may increase the odds of sNDI and motor delay.

## 1. Introduction

Bronchopulmonary dysplasia (BPD) [1] occurs in >40% of infants born <29 weeks gestational age (GA) [2]. It is associated with long-term consequences both in terms of neurodevelopmental challenges and pulmonary function [3,4]. Postnatal steroids either in the systemic (sPNS) or inhaled form have been evaluated in randomized controlled trials for the prevention or treatment of BPD due to their anti-inflammatory properties. Early administration of sPNS (<8 days of life) facilitates extubation and reduces BPD risk. However, short-term adverse effects including gastrointestinal bleeding, intestinal perforation, hyperglycemia, hypertension, and most importantly the long-term increased possibility of cerebral palsy (CP) outweigh the benefits [5]. A recent network meta-analysis evaluating 14 different regimens of corticosteroids, concluded that moderately early (8–14 days) corticosteroids with a cumulative dose of 2–4 mg/kg over < 8 days may be the most effective regimen to prevent mortality or BPD at 36 weeks post-menstrual age (PMA) but was associated with increased risk of hypertension [6]. Importantly, this regimen was not associated with neurodevelopmental impairment (NDI), although, the quality of evidence was low. Due to persisting concerns about CP, several professional guidelines recommend that sPNS should only be administered in preterm infants who cannot be extubated from invasive ventilation in conjunction with parental approval [7,8].

Some of the proposed mechanisms linking steroids to NDI are alterations in the neuronal cell size and number, selective changes in the synaptic function and cell signaling, and neuronal apoptosis [9,10,11]. Various studies have demonstrated that administration of steroids is associated with decreased head growth, and impaired motor and cognitive development [4,12,13,14] when administered early (≤7 days of life), but the evidence of its effects at a specific postnatal age (PNA) beyond 7 days of life is limited and still unclear [15]. Wilson-Costello et al. using data from the National Institute of Child Health and Human Development (NICHD) Neonatal Research Network reported that treatment with sPNS after 33 weeks of PMA was associated with the greatest risk of NDI or death [16] while Harmon et al. [17] reported that the odds for NDI was not significantly different when compared to the reference group (starting steroids between 22-28 days of life) except when initiated between 6–7 weeks of life. Considering the conflicting results, the goal of this study was to establish an association if any, between PNA of sPNS administration for BPD and neurodevelopmental outcomes at 18–24 months corrected age (CA) in preterm infants born at <29 weeks’ GA.

## 2. Methods

### 2.1. Study Design and Population

A retrospective cohort study of preterm infants < 29 weeks GA admitted to the participating neonatal intensive care units (NICUs) in the Canadian Neonatal Network (CNN), between 1 April 2009, and 31 December 2016 was conducted. Infants whose follow-up data were available at 18–24 months CA in the Canadian Neonatal Follow-Up Network (CNFUN) database were included for the assessment of neurodevelopmental outcomes. Infants who were exposed to sPNS after the 1st week of age for BPD were included. We excluded infants who were moribund, had major congenital anomalies, died or received sPNS within the first week of age, and received the first course of sPNS for BPD for a duration of ≤3 or >10 days (Figure 1). The rationale for selecting this window of 3–10 days of sPNS was to align with the DART protocol [18] which is a consistent practice with regard to postnatal dexamethasone use in CNN NICUs. All eligible infants were categorized according to their PNA at first sPNS exposure into 8 groups: weeks 2, 3, 4, 5, 6, 7, 8–9, and ≥10.

### 2.2. Outcomes

The primary outcome was a composite of mortality or significant neurodevelopmental impairment (sNDI) assessed using the Bayley Scales of Infant and Toddler Development-III (BSID-III) at 18–24 months CA. The secondary outcomes included mortality by 18–24 months CA, sNDI, NDI, CP, Bayley III cognitive, language, and motor composite score < 70; and Bayley III cognitive, language, and motor composite score < 85 at 18–24 months CA.

### 2.3. Data Extraction and Definitions

Baseline maternal and neonatal characteristics were abstracted from the CNN database. Data are collected from patient charts at each participating NICUs following procedures outlined in the CNN Abstractor’s Manual [19]. Once the data are entered electronically, they are transmitted to the central coordinating center at Mount Sinai Hospital, Toronto. The CNN database has demonstrated high reliability and internal consistency [20]. Information on the chronological and PMA at first sPNS administration for BPD, length of sPNS, short-term neonatal outcomes, including BPD, severe neurological injury (defined as intraventricular hemorrhage ≥ grade 3 [IVH] or periventricular leukomalacia (PVL) [21], severe retinopathy of prematurity (ROP: stage ≥ 3 or requiring laser treatment) [22], necrotizing enterocolitis (NEC) ≥ stage 2 [23], spontaneous intestinal perforation (SIP), patent ductus arteriosus (PDA), late-onset sepsis, mortality before discharge from NICU and duration of NICU stay were also analyzed.

Mortality after NICU discharge and neurodevelopmental assessment at 18–24 months CA were retrieved from the CNFUN database. The neurodevelopmental assessment included a neurological examination, developmental assessment using the BSID-III, and hearing and visual function as per audiology and ophthalmology report [24]. Gross Motor Function Classification System (GMFCS) was used to determine the level of functional impairment in children with cerebral palsy [25]. sNDI was defined as any one or more of the following: CP with GMFCS scores of 3–5, BSID-III cognitive, language, or motor composite score < 70, the requirement for hearing amplification, or bilateral visual impairment. Children who were untestable on the BSID- III with a BSID-III Adaptive Behaviour score of <70 were classified as having sNDI. NDI was defined as any one or more of the following: CP with a GMFCS score of 1 or higher, a BSID-III cognitive, language, or motor composite score < 85, sensorineural/mixed hearing loss or blindness (unilateral visual impairment), and infants untestable on BSID-III with a BSID-III Adaptive Behaviour score < 85 were classified as NDI.

### 2.4. Statistical Analysis

Maternal and infant characteristics, delivery data and short-term outcomes were evaluated between the sPNS exposed and the no-sPNS exposed groups using the Chi-square test for categorical variables and Student’s *t*-test or Wilcoxon Rank Sum test for continuous variables. To examine the association between infant characteristics and the age at first sPNS, infant characteristics were compared among the age at sPNS groups using the Chi-square test for categorical variables and ANOVA (F test) for continuous variables.

To determine if an association between the long-term outcomes and the age at first sPNS administration was present or not, a non-linear regression analysis using the quadratic model aX^2^ + bX + c was conducted to fit the outcome rates. When the coefficient of the quadratic term was significantly greater than 0 it implied that there was a significant ‘U’ shaped relationship between the rate of outcome and age at sPNS. Multiple logistic regression with quadratic models using a generalized estimating equation (GEE) approach was also conducted to further determine the ‘U’-shaped association between the binary outcomes and age at first sPNS adjusted for potential confounders including GA, small for gestational age (SGA), sex, and CNN site. When no ‘U’ shaped relationship was observed in the adjusted analysis, multivariable linear regressions were conducted to examine the linear relationship between age at first sPNS and neurodevelopmental outcomes. In fitting the models, the symmetric covariance structure was used in the models to account for the clustering of subjects including multiples within a hospital. Data management and statistical analyses were performed using SAS 9.4 (SAS Institute, Inc., Cary, NC, USA) and R 4.0.0. (www.r-project.org, accessed on 20 November 2021). A two-sided *p*-value of < 0.05 was considered statistically significant. 

## 3. Results

A total of 12,806 infants born at <29 weeks were admitted to CNN NICUs, of whom 2358 (18.4%) were excluded during the study duration (Figure 1). Of the remaining 10,448 infants, 6200 infants (59.3%) with available follow-up data were included in the analysis. Maternal and infant characteristics between the sPNS and no-sPNS exposure groups are presented in Table 1. Infants who received sPNS were born at younger GA, had lower birth weight (BW), lower Apgar score < 7 at 5 min of life, were sicker (Score for Neonatal Acute Physiology-II [SNAP-II] > 20), and a higher percentage of them were exposed to antenatal steroids. Severe neurological injury and PDA were higher in the steroid exposed infants. Table 2 reports the comparisons of morbidities between the sPNS exposed and no-sPNS exposed groups. The rate of BPD, severe ROP and late-onset sepsis was higher with a longer duration of NICU stay in infants who received sPNS compared to those who did not receive sPNS. The mortality before discharge from NICU was higher in infants not exposed to steroids. The distribution of the infants according to the age at first sPNS administration was as follows: 8%, 17.5%, 23.1%, 18.7%, 12.6%, 8.3%, 5.8%, and 6% in the 2nd, 3rd, 4th, 5th, 6th, 7th, 8–9th, and ≥10th week of PNA respectively.

The mean age of starting sPNS was 31 days (SD 15.4) and the mean PMA was 30 weeks (SD 2.7). Maternal and infant characteristics amongst the 8 groups, within the steroid exposed infants, starting from week 2 to ≥ week 10 for the sPNS exposure are shown in Table 3. The comparison of outcomes within the infants who received sPNS, among the age groups at the initiation of sPNS for BPD is reported in Table 4. Among the infants assessed for the outcomes, 6 infants had sNDI and subsequent mortality. A linear relationship between initiation of sPNS and a composite of mortality or sNDI, sNDI, CP, BSID-III language and motor composite < 70; and BSID-III motor composite < 85 was noted on univariate analysis as shown in the [Figure 2A–C and Appendix A]. A U-shaped relationship was observed for the outcome of mortality, BSID-III cognitive composite < 70, BSID-III language and cognitive composite < 85; and NDI with the age at initiation of the sPNS on univariate analysis [Figure 2D; Appendix A]. Multiple logistic regression analysis adjusted for the confounders was performed to assess the association of outcomes at 18–24 months CA with PNA at first administration of sPNS (Table 5). There was no significant difference in the primary composite outcome of mortality or sNDI with the PNA at the first administration of sPNS. In addition, there was no significant difference in NDI, CP (GMFCS ≥ 1), or BSID-III composite score < 85 for cognition, language, or motor components. However, there was an increase in the odds of sNDI by 1.5% (95% CI 0.4, 2.9) and BSID-III motor composite < 70 by 2.6% (95% CI 0.9, 4.4) with each one-week increment in the age at initiation of sPNS when compared to the preceding week of initiation. Infants who were lost to follow-up were mostly singleton, had higher GA, had lower SNAP-II scores at the time of admission, and had less severe neurological injury compared to those who were followed up at 18–24 months CA (Appendix A).

## 4. Discussion

A “safe window” of PNA for the use of sPNS for BPD, with minimal risk for NDI, remains elusive. In this national study, no difference in the composite outcome of mortality or sNDI and PNA at first administration of sPNS for BPD in preterm infants < 29 weeks GA was noted. We observed that approximately 60% of infants received PNS between the 3rd and 5th week of PNA. Even though the composite outcome was not different, amongst survivors, the odds of sNDI were 1.5%, and Bayley-III motor composite < 70 was 2.6% higher with each week’s delay in the initiation of sPNS when compared to the preceding week, commencing from week 2 of PNA.

Many studies [4,12,13,14] previously have evaluated the effect of sPNS used for BPD on neurodevelopmental outcomes in infants born preterm. However, there are only 2 studies published to date that have addressed the question of the association of the risk of NDI with the timing of exposure to sPNS. Wilson-Costello et al. [16] reported that each 1 mg/kg steroid exposure was associated with a 2.0-point decrease in mental development index and 40% increased odds of CP [odds ratio (OR) 1.4; 95% CI 1.2, 1.6] and that the possibility of CP was greater in the steroid exposed infants at every PNA evaluated compared to unexposed infants.

Among the steroid-exposed infants, the occurrence of NDI/death and CP was higher within all the total steroid dose tertiles-compared to those not exposed to sPNS. They observed that infants with older PMA at the time of exposure, despite not receiving the highest total dose had a greater possibility of NDI. Wilson-Costello et al. also noted that sPNS treatment after 33 weeks of PMA was associated with the greatest risk of NDI/death. This observation was similar to our finding of a higher likelihood of NDI with introduction of sPNS at later PNA. In the study by Harmon et al. [17] the median age of initiation of PNS was 31 days, which was comparable to our study. They concluded that treatment with sPNS initiated between 2- and 7-weeks of PNA for BPD was not associated with NDI. However, the safety window reported by the authors was wide. Despite our attempt, we were unable to identify an “optimal safe PNA window” for the introduction of sPNS with regard to the composite outcome of mortality or sNDI. We noted that there was a linear univariable relationship between the age at initiation of sPNS and sNDI or Bayley-III motor composite < 70. Contradictory to the existing evidence, a more recent network meta-analysis [6] of various steroid regimens either for prevention or treatment of BPD on long-term outcomes indicated that a high cumulative dose of dexamethasone (>4 mg/kg) had a lower incidence of NDI compared with medium cumulative dose (2–4 mg/kg) when administered between 15–27 days of life. Thus, evidence to identify the safe period for initiation of sPNS for BPD has been tenuous.

Our observation of no difference in the primary outcome could potentially be explained by survivor bias in the steroid exposed group. Mortality alone was observed to have a U-shaped relationship while sNDI alone had a linear relationship with the PNA at the introduction of the systemic steroids for BPD producing no effect on the composite outcome. The finding of increased risk of sNDI or Bayley-III motor composite < 70 with the initiation of sPNS at later PNA may be explained by the theory that the deleterious neurodevelopmental effects of steroids may be additive to the effects resulting from ongoing intermittent hypoxia due to prolonged mechanical ventilation coupled with evolving lung disease on the developing brain via lung-brain axis. The excitotoxicity related to glutamate in hypoxic-ischemic induced brain injury combined with the heightened excitotoxicity produced by dexamethasone amplifies the risk of neuronal insult [26]. The stage of brain development at the time and duration of steroid exposure determines its impact. Exposure to steroids decreases the affinity of the N-methyl D-aspartate (NMDA) receptor [10] which may affect synaptic plasticity, learning, and memory functions. This effect is pronounced in early postnatal life, but not in fetuses or adults, indicating a particular period of vulnerability to the effects of steroids [10]. Though these effects were observed in a preclinical context, there are no clinical studies to support these findings except for the observation of higher sNDI with increasing PNA/PMA as noted in this study and Costello et al. [16].

The key strength of this study is the use of a national cohort of infants < 29 weeks GA. We acknowledge several limitations of our study. First, follow-up data were missing in 41% of our population. However, infants without follow-up data were of higher GA, had lower SNAP-II scores and had less severe neurological injury. Second, by excluding the infants who received sPNS for ≤3 or >10 days, some infants with the primary outcome may have been missed, but our goal was to evaluate the effect of the timing of exposure to the first steroid course of 10 days. Third, the cumulative dose of the first course of sPNS and the type of steroid administered is not available in the CNN database, therefore an association between the dose and type of steroid and outcomes could not be ascertained. Fourth, we did not evaluate the effect of early prolonged hydrocortisone on the NDI, as hydrocortisone was not widely used during the study period in participating NICUs. The PREMILOC study [27] was published in April 2016, which is during the last 8 months of our study duration (1 April 2009, to 31 December 2016), hence the use of hydrocortisone in Canadian centers was limited. Fifth, the effect of multiple courses of sPNS and its association with the composite outcome was not evaluated. Lastly, we anticipate that there is variation in neonatal intensive care practices across all the units within the network which can have an impact on both short- and long-term outcomes irrespective of systemic steroid use.

Though the effects of sPNS have always been a concern from a neurodevelopmental perspective in preterm neonates, each NICU has a different approach to deciding the type of steroid use (dexamethasone versus hydrocortisone) and the age of initiation of sPNS. Our findings support the evidence that the age at the introduction of sPNS for BPD does not affect the composite outcome of mortality or sNDI. Further studies, including infants exposed to hydrocortisone and inhaled steroids and predicting the probability of BPD using the BPD risk calculator to maximize the pulmonary effects and minimize the CNS effects of systemic steroids, are required.

## 5. Conclusions

The postnatal age of initiation of sPNS for BPD had no impact on mortality or sNDI among Canadian preterm infants. Among survivors, each week’s delay in initiation of sPNS may be associated with increased odds of sNDI and motor delay.

## Figures and Tables

**Figure 1 children-09-01687-f001:**
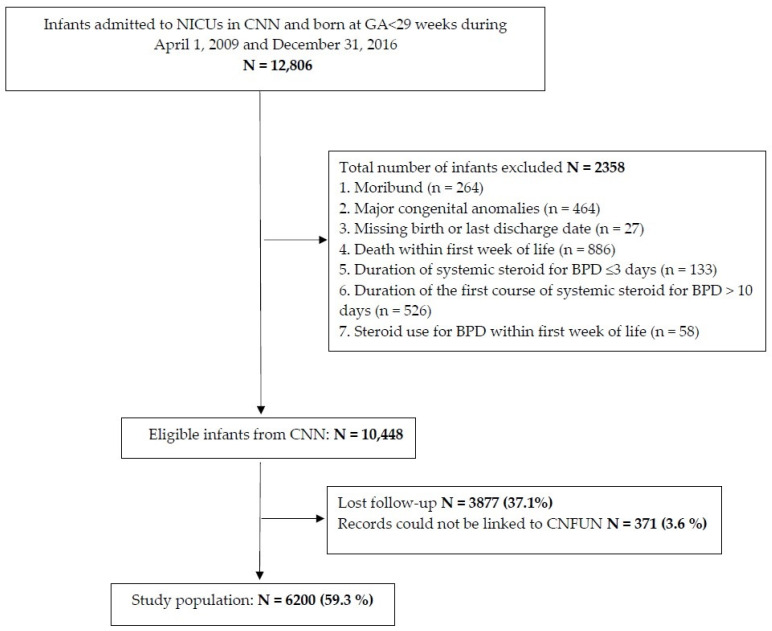
Study population flow chart.

**Figure 2 children-09-01687-f002:**
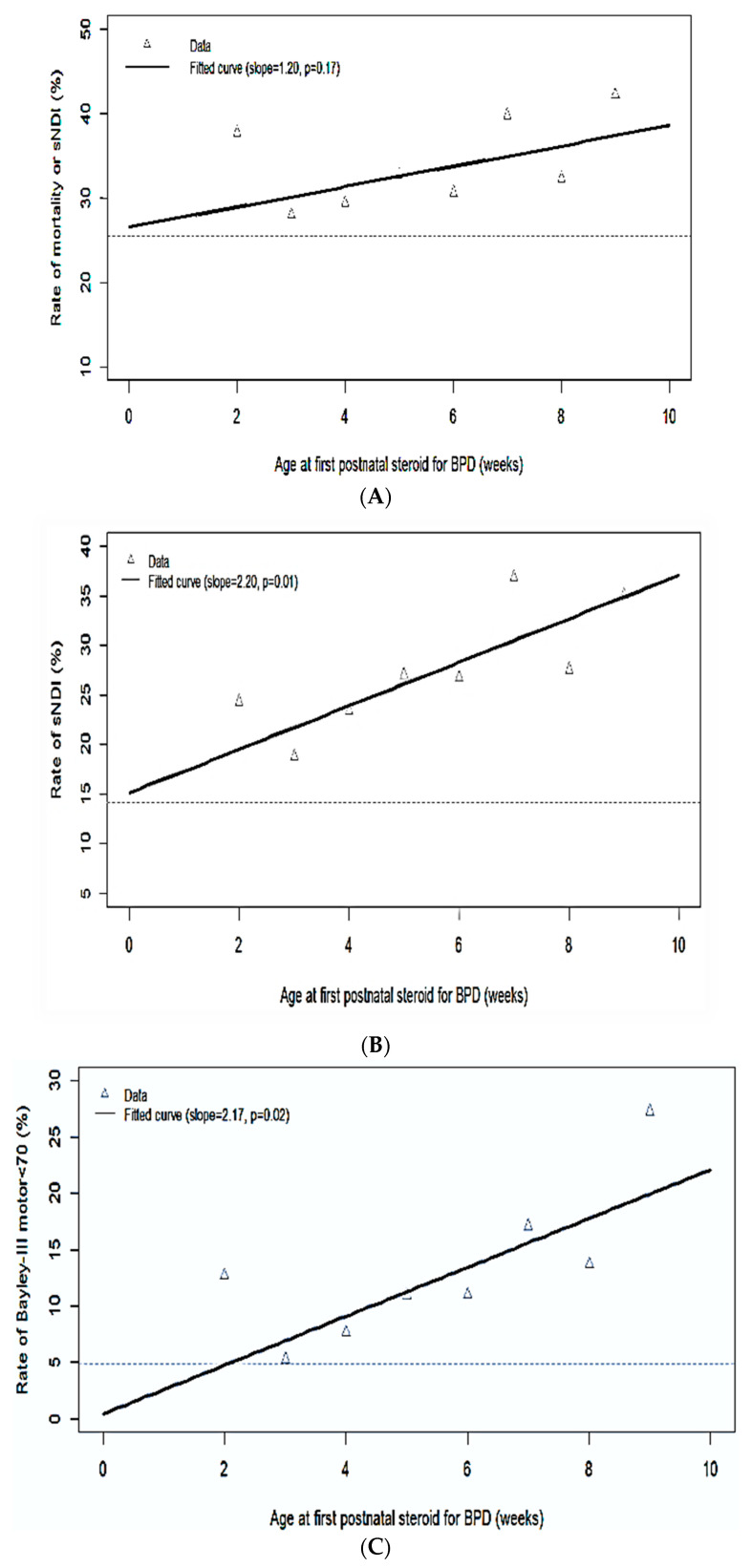
Relationship of the outcomes with the postnatal age of starting sPNS on the unadjusted univariate analysis. A linear relationship between initiation of sPNS and a composite of mortality or sNDI, sNDI, motor composite < 70 (**A**–**C**) while a U-shaped relationship for mortality (**D**) is noted.

**Table 1 children-09-01687-t001:** Maternal and infant characteristics based on exposure to systemic postnatal steroids.

Variables *	No sPNS Exposure (N = 5137)	sPNS Exposure (N = 1063)	*p*-Value
Maternal age, mean (SD)	31.2 (5.82)	31.3 (5.71)	0.7
Maternal hypertension, % (*n*/N)	1.9 (800/5008)	17.5 (184/1050)	0.22
Maternal diabetes, % (*n*/N)	9.6 (472/4904)	7.8 (79/1014)	0.07
Antenatal steroid use, % (*n*/N)	89.2 (4486/5030)	91.6 (958/1046)	0.02
Cesarean delivery, % (*n*/N)	58 (2971/5120)	58.3 (618/1061)	0.89
Singleton, % (*n*/N)	72.5 (3722/5135)	74.4 (791/1063)	0.2
Gestational age at birth (weeks), median (IQR)	27 (25, 28)	25 (24, 26)	<0.001
Gestational age group, % (*n*/N) 22–25 (weeks) 26–28 (weeks)	25.8 (1323/5137) 74.2 (3814/5137)	56 (595/1063) 44 (468/1063)	<0.001
Birth weight (g), mean (SD)	950 (234)	799 (202)	<0.001
Sex (male), % (*n*/N)	53.4 (2742/5132)	55.6 (589/1060)	0.2
Small gestational age, % (*n*/N)	8 (411/5134)	10.7 (113/1061)	0.005
Apgar score < 7 at 5 min, % (*n*/N)	38.1 (1938/5087)	45.1 (474/1052)	<0.001
SNAP-II score > 20, % (*n*/N)	26.5 (1354/5109)	36.8 (390/1061)	<0.001
Severe neurological injury (≥grade III IVH or PVL), % (*n*/N)	12.9 (646/5008)	15.9 (166/1038)	0.008
PDA, % (*n*/N)	55.1 (2815/5106)	71.9 (759/1055)	0.001

* Data are presented as mean (SD), median (IQR) or % (*n*/N) as appropriate, IQR = inter-quartile range; IVH = intraventricular hemorrhage; N = number; PDA = Patent ductus arteriosus; PVL = Periventricular leucomalacia; SD = standard deviation; SNAP-II = Score for Neonatal Acute Physiology-II; sPNS = systemic postnatal steroid.

**Table 2 children-09-01687-t002:** Short-term outcomes of infants based on exposure to systemic postnatal steroids.

Variables *	No sPNS Exposure (N = 5137)	sPNS Exposure (N = 1063)	*p*-Value
BPD, % (*n*/N)	44 (1968/4472)	63.4 (637/1004)	<0.001
Severe ROP (≥stage III or requiring therapy), % (*n*/N)	9 (333/3483)	24 (236/978)	<0.001
NEC stage ≥ 2, % (*n*/N)	9.6 (493/5127)	8 (87/1062)	0.15
SIP, % (*n*/N)	3 (138/4568)	4 (37/964)	0.19
Late-onset sepsis, %(*n*/N)	24.6 (1263/5137)	34.6 (368/1063)	<0.001
Mortality before discharge from NICUs, %(*n*/N)	13.1 (674/5137)	9 (92/1063)	<0.001
Duration of NICU stay (days), median (IQR)	64 (36, 91)	104 (65, 128)	<0.001

* Data are presented as median (IQR) or % (*n*/N) as appropriate, BPD = bronchopulmonary dysplasia; IQR = inter-quartile range; NEC = necrotizing enterocolitis; NICU = neonatal intensive care unit; N = number, ROP = retinopathy of prematurity; SIP = spontaneous intestinal perforation; sPNS = systemic postnatal steroid.

**Table 3 children-09-01687-t003:** Maternal and infant characteristics among the infants exposed to steroids.

Variables *	Postnatal Age at Exposure to the First sPNS (for BPD)							
	Week 2 N = 85	Week 3 N = 186	Week 4 N = 245	Week 5 N = 199	Week 6 N = 134	Week 7 N = 88	Week 8–9 N = 62	≥Week 10 N = 64
Maternal hypertension, % (*n*/N)	12 (10/84)	19 (34/184)	17 (42/241)	16 (32/199)	16 (21/130)	16 (14/87)	14 (9/62)	35 (22/63)
Antenatal steroid use, % (*n*/N)	84 (71/84)	92 (165/180)	93 (222/239)	93 (185/199)	92 (121/132)	84 (72/86)	98 (61/62)	95 (61/64)
Cesarean delivery % (*n*/N)	67 (57/85)	57 (106/186)	59 (145/244)	56 (111/199)	56 (75/133)	57 (50/88)	56 (35/62)	61 (39/64)
Singleton, % (*n*/N)	80 (68/85)	79 (147/186)	71 (174/245)	76.4 (152/199)	72.4 (97/134)	70.5 (62/88)	64.5 (40/62)	79.7 (51/64)
Gestational age at birth (weeks) median (IQR)	25 (24, 26)	25 (24, 26)	25 (24, 26)	25 (25, 26)	25 (24, 26)	25 (24, 26)	26 (24, 27)	26 (25, 27)
Gestational age group, % (*n*/N) 22–25 (weeks) 26–28 (weeks)	68 (58/85) 32 (27/85)	57 (106/186) 43 (80/186)	57 (139/245) 43 (106/245)	55 (110/199) 45 (89/199)	54 (72/134) 46 (62/134)	57 (50/88) 43 (38/88)	47 (29/62) 53 (33/62)	48 (31/64) 52 (33/64)
Birth Weight (g), mean (SD)	780 (187.18)	814 (184.43)	802 (187.59)	786 (222.47)	822 (185.21)	791 (262.86)	819 (211.99)	756 (176.8)
Sex (male), % (*n*/N)	65 (55/85)	55 (103/186)	58 (142/243)	52 (103/199)	54 (73/134)	51 (44/87)	55 (34/62)	55 (35/64)
Small gestational age, % (*n*/N)	11 (9/85)	7 (13/186)	9 (23/243)	13 (25/199)	9 (12/134)	12 (11/88)	8 (5/62)	23 (15/64)
Apgar score < 7 at 5 min, % (*n*/N)	52 (44/84)	48 (89/184)	40 (97/243)	41 (82/199)	47 (61/131)	38 (33/86)	56 (35/62)	52 (33/63)
SNAP-II score > 20, % (*n*/N)	43 (36/84)	29 (54/186)	42 (102/244)	34 (67/199)	33 (44/134)	39 (34/88)	44 (27/62)	41 (26/64)

* Data are presented as mean (SD), median (IQR) or % (*n*/N) as appropriate, BPD = bronchopulmonary dysplasia; IQR = inter-quartile range; N = number; SD = standard deviation; SNAP-II = Score for Neonatal Acute Physiology-II; sPNS = systemic postnatal steroid.

**Table 4 children-09-01687-t004:** Neurodevelopmental outcomes in the infants exposed to systemic postnatal steroids.

Outcomes *	Age at First sPNS for BPD							
	Week 2 N = 85	Week 3 N =186	Week 4 N = 245	Week 5 N = 199	Week 6 N = 134	Week 7 N = 88	Week 8–9 N = 62	≥Week 10 N = 64
Mortality or sNDI, % (*n*/N)	38 (32/85)	28 (52/186)	29 (72/245)	33 (65/199)	31 (41/134)	40 (35/88)	32 (20/62)	42 (27/64)
Mortality, % (*n*/N)	18 (15/85)	11 (21/186)	9 (21/245)	9 (17/199)	6 (8/134)	5 (4/88)	8 (5/62)	11 (7/64)
sNDI, % (*n*/N)	24 (17/70)	19 (31/165)	24 (53/226)	27 (50/185)	27 (34/127)	37 (31/84)	28 (16/58)	35 (20/57)
NDI, % (*n*/N)	60 (42/70)	55 (90/165)	53 (119/226)	52 (96/185)	55 (70/127)	61 (51/84)	62 (36/58)	61 (35/57)
Any CP (GMFCS ≥ 1), % (*n*/N)	9 (6/67)	8 (12/161)	7 (16/221)	8 (15/179)	9 (11/124)	11 (9/83)	9 (5/55)	13 (7/55)
Cognitive composite < 85, % (*n*/N)	25 (16/65)	21 (33/155)	20 (41/205)	19 (33/175)	21 (25/120)	32 (24/76)	25 (13/53)	35 (16/46)
Cognitive composite < 70, % (*n*/N)	9 (6/65)	5 (8/155)	4 (8/205)	6 (10/175)	7 (8/120)	7 (5/76)	8 (4/53)	13 (6/46)
Language composite < 85, % (*n*/N)	48 (31/64)	45 (67/149)	44 (87/200)	40 (71/176)	41 (48/117)	49 (36/73)	49 (24/49)	58 (26/45)
Language composite < 70, % (*n*/N)	22 (14/64)	13 (19/149)	17 (34/200)	17 (30/176)	17 (20/117)	27 (20/73)	20 (10/49)	29 (13/45)
Motor composite < 85, % (*n*/N)	35 (22/63)	25 (38/152)	25 (49/195)	27 (47/173)	33 (39/118)	41 (31/76)	31 (16/51)	48 (21/44)
Motor composite < 70, % (*n*/N)	13 (8/63)	5 (8/152)	8 (15/195)	11 (19/173)	11 (13/118)	17 (13/76)	14 (7/51)	27 (12/44)

* Data are presented as % (*n*/N), CP = cerebral Palsy; GMFCS = Gross Motor Functional Classification; NDI = neurodevelopmental impairment; sNDI = significant neurodevelopmental impairment.

**Table 5 children-09-01687-t005:** Adjusted odds of neurodevelopmental outcomes for the age at first systemic postnatal steroid use.

Outcomes *	Adjusted OR * (95% CI) (Per 1-Week Increase in the Postnatal Age)	*p*-Value
Mortality or sNDI	1.094 (0.942, 1.024)	0.22
sNDI	1.015 (1.004, 1.029)	0.04
NDI	1.007 (0.995, 1.019)	0.25
CP (GMFCS ≥ 1)	1.01 (0.99, 1.03)	0.35
Bayley III cognitive composite < 85	1.009 (0.999, 1.019)	0.05
Bayley III language composite < 85	1.006 (0.986 1.013)	0.11
Bayley III motor composite < 85	1.015 (0.999, 1.030)	0.06
Bayley III cognitive composite < 70	1.013 (0.995, 1.033)	0.15
Bayley III language composite < 70	1.012 (0.997, 1.027)	0.11
Bayley III motor composite < 70	1.026 (1.009, 1.044)	0.002

* Adjusted OR were based on the multivariable logistic regression adjusted for GA, SGA, sex using the GEE approach to account for the clustering of infants within the CNN site, CNN = Canadian neonatal network; CI = confidence interval; CP = cerebral palsy; GA = gestational age; GEE = generalized estimating equation; GMFCS = gross motor functional. classification system; NDI = neurodevelopmental impairment; OR = odd ratio; sNDI = significant neurodevelopmental impairment; SGA = small for gestational age.

## Data Availability

Deidentified individual participant data will not be made available.

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
