# Peer review of "Timing of Systemic Steroids and Neurodevelopmental Outcomes in Infants < 29 Weeks Gestation"

_children, 2022, doi:10.3390/children9111687_

Round 1

Reviewer 1 Report

This is a very well written manuscript and an important study to understand the association of postnatal steoid use and its impact on mortality and neurodevelopmental outcome. This question even more relevant now as we are managing infants of smaller gestational ages on our neonatal intensive care units.

In this large, population-based, retrospective cohort study, the authors have concluded that there is no significant difference in the composite outcome of mortality and neurodevelopmental outcome with use of postnatal systemic steroids. The other interesting finding which authors have mentioned is that amongst the survivors, the odds of sNDI were 1.5%, and Bayley-III motor composite <70 was 2.6% higher with each week's delay in initiating postnatal systemic steroids.

There is a large sample size in this study. The authors were unable to identify the optimal safe postnatal age to start systemic steroids.

The study has several limitations. This is a retrospective study. Quite a number of infants were lost over follow up. he authors though have justified that these infants were of larger gestation and less severe. Thirdly, it is not clear if all units across the network follow the same protocols on managing infants of all gestations. The authors need to clearly mention the variability in neonatal intensive care practices across all the units within the network.

The study includes data from 1 April 2009 to 31 December 2016. The authors should state if there were any major changes to the practices during any of these time periods. There is no mention of the type of systemic steroids used by the units and this could have an impact on the results.

With the clarification of the above queries, the manuscript is good to be published. It will add more evidence to the current knowledge on use of systemic postnatal steroids for BPD.

Reviewer 2 Report

This is a valuable study to investigate the relationship between systemic postnatal steroids and neurodevelopmental outcomes.  However, there are some comments as follows:

1.     Were the systemic steroids used in the hospitals of the Canadian Neonatal Network to prevent bronchopulmonary dysplasia (BPD) all dexamethasone? Or were there other steroids such as hydrocortisone used in NICUs as drugs for the treatment of BPD? Did different systemic steroids have different outcomes?

2.     Furthermore, according to the DART study dosing regimen, the therapeutic course of dexamethasone is 10 days. The regimens that is included in this article included 3 to10-day therapeutic courses. Did these different total cumulative doses of steroids affect the outcomes?

3.     Was there any case receiving a second systemic steroid treatment course? In addition to the initiation timing of treatment, should the number of treatments also be considered for analysis?
